# The Pattern of Gene Expression (*Igf* Family, Muscle Growth Regulatory Factors, and Osteogenesis-Related Genes) Involved in the Growth of Skeletal Muscle in Pikeperch (*Sander lucioperca*) During Ontogenesis

**DOI:** 10.3390/ani14213089

**Published:** 2024-10-26

**Authors:** Fatemeh Lavajoo Bolgouri, Bahram Falahatkar, Miquel Perelló-Amorós, Fatemeh Moshayedi, Iraj Efatpanah, Joaquim Gutiérrez

**Affiliations:** 1Fisheries Department, Faculty of Natural Resources, University of Guilan, Sowmeh Sara, 1144, Guilan, Iran; f.lavajoo@gmail.com (F.L.B.); iefatpanah@yahoo.com (I.E.); 2Department of Cell Biology, Physiology and Immunology, Faculty of Biology, University of Barcelona, 08028 Barcelona, Spain; miquelperelloamoros@gmail.com (M.P.-A.); fateme.moshayedi@yahoo.com (F.M.)

**Keywords:** pikeperch, gene expression, larval development, regulatory metabolism, aquaculture

## Abstract

This study focused on the mRNA expression of various growth-related and osteogenesis genes in pikeperch (*Sander lucioperca*) during its developmental stages from hatching to 40 days post-hatching (DPH). The average total length of the larvae increased from 3.6 mm at hatching to 27.1 mm by 40 DPH. Results showed distinct phases of gene expression across the egg, larval, and juvenile stages, indicating a transition toward juvenile development. According to the PLS-DA model, the most relevant VIPs are myf5 and *mymk* as best markers of earlier stages and *igf1ra*, *ostc*, *pax7*, and *ghr* as markers of later stages of ontogeny. Overall, these findings highlight the dynamic changes in gene expression that regulate metabolism, growth, and development in pikeperch, providing insights important for pikeperch farming.

## 1. Introduction

The pikeperch (*Sander lucioperca*) is a member of the Percidae family. A large, diverse, and economically important group of primarily freshwater fishes, it includes 11 genera and about 275 known species [1]. The rapid growth trait of the pikeperch relative to other Percidae and its potential for diversification make it an attractive species for intensive rearing.

The growth rate of fish during the early life stage can vary depending on the stages. The pikeperch developmental stages consist of embryonic organ formation, hatching, and the transition from endogenous to exogenous feeding in parallel to acquiring of many new physiological functions [2,3,4]. According to the FAO [5], the constant decline in the annual capture of pikeperch is estimated from 48,800 tones (1950s) to 21,200 tones (nowadays). Thus, pikeperch aquaculture in artificial conditions has developed due to the increasing demand for this species for human consumption [6]. Improving the quality and quantity of the myotomal muscle is an essential objective in the aquaculture industry. However, different problems affect the increased production and trade in aquaculture, and in pikeperch, it is hampered by difficulties, especially during early ontogenesis, which includes the organo-, myo-, skeleto-, and neurogenesis, as well as the phase of growth and the development of the immune system [7]. So, basic knowledge of the physiological processes during rearing is important for improving aquaculture production. Hence, studying the expression of the growth, muscle, and bone gene pattern is necessary during the ontogeny stages.

The implication of the endocrine system in regulating teleost’s growth during early development is obvious, and information is available [8,9]. Like other vertebrates, the primary regulator of the somatic growth and development of fish is carried out by the *growth hormone*/insulin-like growth factor axis [10,11]. Both growth hormone/*insulin*-like growth factors are crucial during early fish development and differentiation [10,12]. The formation of skeletal muscular tissue is initiated in fish embryos during early development compared with birds and mammals [8,13,14], and this dynamic and plastic tissue contributes to the fish swimming force at early life stages. This is because fish embryos develop in an aqueous environment, requiring functional musculature at an earlier stage to facilitate movement and survival. During embryogenesis, skeletal muscle is formed by cells derived from somites that differentiate into a myotome [8]. The differentiation of somites into different types of the muscle depends on specific and extreme functions of myogenic regulatory factors (*mrfs*) [15]. Moreover, *mrfs* control the determination and differentiation of skeletal muscle cells during embryogenesis and postnatal myogenesis [8,16]. In addition to muscles, bones play an essential role in fish locomotion. The osteogenesis process is regulated by skeletal-derived factors that control specific stages of osteoblast development and bone building [17]. Moreover, synchronicity between bone and muscle is required for proper musculoskeletal growth [18].

There is little information concerning regulating growth, muscle, and bone molecular factors during pikeperch ontogeny. However, Franz et al. [14] studied the regulation of myogenic genes during the embryo–larval transition in pikeperch for the first time. Therefore, our main objective was to evaluate the gene expression pattern of the growth hormone/insulin-like growth factor axis, muscle growth regulatory factors, and osteogenesis-related genes involved in the growth of the skeletal muscle in pikeperch under culture conditions. Our data can be useful for exploring basic knowledge about pikeperch growth during early ontogenesis for sustainable aquaculture.

## 2. Materials and Methods

### 2.1. Larval and Juvenile Rearing

The larvae used in the present study were obtained by spontaneous spawning of pikeperch broodstock held at controlled conditions under optimal temperature 13.9 ± 0.4 °C with high fertilization, survival, and hatching rates. This study was conducted in the Dr. Yousefpour Marine Fishes Restocking and Genetic Conservation Center (YFHC, Siahkal, Guilan, Iran). Pikeperch wild broodstock was captured from the lake behind the Aras dam in northwest Iran and transported to the YFHC. The age and body weight of caught fish were 4–5 years and 1.1 ± 0.1 kg, respectively. Before the spawning, breeders were transferred to twelve circular concrete tanks (185 cm diameter × 40 cm depth) without feeding for 5–7 days. For hormonal induction, fourteen females and sixteen males were held in each rectangular concrete tank (13 × 3.08 × 1.1 m) with an artificial spawning nest (50 × 50 cm) [19] for each pair. Spawning in female fish was induced by injection of 200 IU kg^−1^ human chorionic gonadotropin (hCG, manufactured by LG life sciences, South Korea), whereas male fish received no injections. At 80–85 h after injection, females spawned on artificial nests. The spawned adhesive eggs attached to the nests were transferred to a circular concrete tank with 180 cm diameter, 50 cm height, and 1272 L of water volume. The water flow was maintained at 0.3–0.5 L min^−1^. The larval density in each circular concrete tank was 70 ind L^−1^. The water physicochemical properties during the experimental period, including temperature, dissolved oxygen, and pH, were 18.7 ± 0.3 °C, 7.2 ± 0.6 mg L^−1^, and 7.8 ± 0.2, respectively. The water ammonia nitrogen was measured under 0.03 mg L^−1^. After hatching, artificial nests were removed from the circular concrete tank to supply enough space for the growth and motion of larvae. Larvae showed swim-up behavior after 2–3 days post-hatch (DPH). The air blower supplied enough oxygen needed by larvae in the circular concrete tank. Rotifers, cyclopoid copepods, copepod nauplii, or some small cladocerans were mainly used as the first prey for the 5th DPH larvae [20,21]. The larvae fed were composed of *Artemia* nauplii and then with *Daphnia magna* at the beginning of the inflection stage (10th DPH), of which the highest average density was kept at 30 ind mL^−1^ three times per day (at 8.00, 13.00, and 18.00) (Figure 1). The small zooplanktons were obtained from a stock produced in an earthen pond (using cow manure as an organic fertilizer for microalgae growth on which the zooplankton feed) [22]. The *Artemia* nauplii was hatched based on the standard procedure [23]. After 16–24 h, about 70% of the cysts at 29 °C were hatched and fed to the larvae. From 15 DPH to the juvenile period, larvae were reared in three earthen ponds under identical conditions including water exchange (5–20% every day), feeding (natural zooplankton such as rotifers, cladocerans, and copepods) [24], water temperature (21.5 ± 2.5 °C), water ammonia nitrogen (controlled under 0.03 mg L^−1^), and natural photoperiod. The pond size was 4 ha, and the larval density was 400 × 10^3^ larva ha^−1^ in the earthen pond. During the experiment, random samples of the eggs before hatch and larvae during 1, 3, 5, 8, 10, 14, and 21 DPH and in the juvenile stage were taken at the same time of the day from 9:00 to 11:00 AM (150 to 200 fish in each stage). Larvae were killed with an overdose of clove powder extract. After total length measurement, the whole fish was immediately frozen in liquid nitrogen. Then, all samples were stored at −80 °C until further analyses.

### 2.2. Ethics Approval

The experimental basic principles of the ARRIVE guidelines approved all the experimental procedures involved in this study. The reporting performed in the manuscript follows the recommendations given in the ARRIVE guidelines for Reporting Animal Research. No distress or suffering was produced by the procedures used to perform this study.

### 2.3. Gene Selection and Primer Design

Altogether, 23 genes were selected for analysis (Table 1). The focus was placed on growth genes: growth hormone receptor (*ghr*), insulin-like growth factor 1 (*igfI*), insulin-like growth factor II (*igfII*), insulin-like growth factor binding protein 4 (*igfbp4*), insulin-like growth factor binding protein 5A (*igfbp5A*), insulin-like growth factor 1a receptor (*igf1ra*), and insulin-like growth factor 1b receptor (*igf1rb*); muscle genes: paired Box 7 (*pax7*), myogenic factor 5 (*myf5*), myogenic differentiation 1 (*myod1*), myogenic differentiation 2 (*myod2*), myogenin (*myog*), myogenic regulatory factors (*mrf4*), myomaker (*mymk*), and myostatin (*mstnb*); bone genes: collagen type I alpha 1 (*col1a1a*), *fibronectin* 1a (*fib1a*), osteocalcin (*ostc*), osteopontin (*op*), and osteonectin (*ostn*); and reference genes: beta-actin (*β-actin*), elongation factor1a (*ef1a*), and ribosomal protein *S18*. The newly designed specific primers for this study were obtained with the NCBI Primer-Blast Tool using as templates the cDNA sequences of the studied genes obtained from the pikeperch genome deposited in the Ensembl (SLUC_FBN_1) and corroborated with reciprocal BLAST against the NCBI nucleotide database. The primer sequence quality was assessed with NetPrimer online software (http://www.premierbiosoft.com/netprimer/netprlaunch/netprlaunch.html accessed on 8 May 2022).

### 2.4. RNA Extraction, cDNA Synthesis, and qPCR Analysis

All the analyses were performed at the Department of Cell Biology, Physiology, and Immunology at the University de Barcelona. Due to the different amounts of sample available, the number of samples for each stage was given as follows: 0 day (egg): 10 samples; day 1: 5 samples; day 3: 10 samples; day 5: 10 samples; day 8: 10 samples; day 14: 10 samples; day 21: 6 samples; Juveniles: 10 samples. For RNA extraction, 1 mL of TRI Reagent Solution^®^ (Applied Biosystems, Alcobendas, Spain) was added to the samples (100 mg of whole pooled larvae for each stage or 100 mg of freeze-powdered whole-body homogenate of the juvenile individuals). The samples were homogenized (5–10 pools for each stage) with the Precellys Evolution^®^ (Bertin Instruments, Montigny-le-Brettoneux, France) [14], coupled with a Cryolys system^®^ (Bertin Instruments, Montigny-le-Brettoneux, France), adjusting the protocol depending on the hardness and elasticity of the tissue. The RNA extraction was performed using TRI Reagent Solution^®^ following the manufacturer’s instructions. Each sample final total RNA concentration was obtained using Nanodrop 2000 TM (Thermo Scientific, Alcobendas, Spain). RNA integrity was confirmed in 1% agarose gel (*m/v*) stained with SYBR-Safe DNA Gel Stain^®^ (Life Technologies, Alcobendas, Spain). Following the manufacturer’s recommendations, reverse transcription was carried out with a First Strand cDNA Synthesis Transcriptor Kit^®^ (Roche, Sant Cugat del Valles, Spain). The expression of each gene was calculated using the Pfaffl method [25], which accounts for the efficiency of each primer pair relative to the geometric mean of the reference genes *βactin*, *Ef1a*, and *rps18*. According to the requirements of the MIQE guidelines [26], the mRNA transcript levels of the genes were analyzed by qPCR using the CFX384 TM RealTime System (Bio-Rad, El Prat de Llobregat, Spain). The expression level of each analyzed gene was calculated relative to the reference genes *βactin*, *Ef1a*, and *rps18* using the Pfaffl method [25]. The analysis was performed in a final volume of 5 µL, containing 2.5 µL of iTaq SYBR Green Supermix^®^ (Bio-Rad, El Prat de Llobregat, Spain), 0.125 µL of forward (250 nM) and reverse (250 nM) primers, 1 µL of cDNA from each sample, and 1.25 µL of DEPC water. The reaction was performed in triplicate in 384-well plates (Bio-Rad, El Prat de Llobregat, Spain) under the conditions described by Salmerón et al. [27]. The qPCR consisted of the following: (1) an activation phase of 3 min at 95 °C; (2) 40 cycles of 10 s at 95 °C and 30 s at 55–68 °C (dependent on the melting temperature of the primers (Table 1)); and (3) a melting curve from 55 °C to 95 °C that increased by 0.5 °C every 30 s. Before this analysis, the adequate cDNA dilution for each gene was determined by a dilution factor with a pool of samples. With this efficiency curves, the specificity of the amplification, the absence of primers dimers, and the efficiency of the primers were also tested.

### 2.5. Statistical Analyses

Data were analyzed using IBM SPSS Statistics v.25 (Armonk, NY, USA) and were presented as means ± standard error of the mean (SEM). The normal distribution was analyzed using the Shapiro–Wilk test, and the homogeneity of the variances (homoscedasticity) was assessed with Levene’s test. If normal distribution and/or homoscedasticity was not found, data were logarithmically transformed. Significant differences were tested by one-way analysis of variance (ANOVA) and the post-hoc Tukey HSD. The nonparametric Kruskal–Wallis test and the post-hoc Games–Howell were used if necessary. Statistical differences were considered significant when *p <* 0.05.

According to published papers [28,29], the peak intensity tables of all genes were uploaded to the websites of MetaboAnalyst 6.0 (https://www.metaboanalyst.ca/MetaboAnalyst/ModuleView.xhtml accessed on 3 April 2024) for data processing and analyses. In the multivariate analysis module of MetaboAnalyst, the normalized data were then subject to partial least squares discriminant analysis (PLS-DA) for pattern discovery.

For the relative expression graphs and the PLS-DA, the gene expression value of each individual biological sample for a given gene was the average of the triplicates (technical replicates of the qPCR). 

## 3. Results

From hatching (67 degree days) to the final period of development (40 DPH, 777 degree days), the total length of the pikeperch continuously grew and increased from 3.6 ± 0.4 mm to 27.0 ± 1.1 mm, respectively (Figure 1).

### 3.1. Growth Hormone Receptor and Igfs Genes Expression

*ghr* gene expression showed a significant increase at 1 DPH and then decreased, presenting the lowest values at 21 DPH and in the juveniles (Figure 2a). Regarding *igfI,* the expression in the egg stage was very low, increasing first after hatching, and then it showed a significant increase at 10 and 14 DPH, returning to the lower levels at the juvenile stage (Figure 2b). On the other hand, *igfII* interestingly presented a relatively high expression in the egg stage, which slightly decreased from 3 DPH to 8 DPH as well as days 10 and 14, similar to *igfI* (Figure 2c). The expression of *igf1bp4* was low in the egg but peaked at 1 and 14 DPH; afterward, its expression diminished in the juveniles (Figure 2d). For *igf1bp5*, a fluctuating profile of expression was observed with peaks on days 1, 5, and 14 DPH, followed by a decrease at 21 DPH and in the juveniles (Figure 2e). Concerning the *igf1* receptors, *igf1ra* gene expression showed a low basal expression at the egg stage, progressively increasing with significant values at 21 DPH and in the juveniles (Figure 2f). The expression of the *igf1rb* (Figure 2g) remained high from 1 to 14 DPH, but very low expression was detected in the egg, 21 DPH, and in the juvenile.

### 3.2. Muscle Responses during Ontogenesis

Regarding *pax7*, the initial expression was very low, which increased at 3 DPH, keeping stable levels until 21 DPH and at the juvenile stage, where it presented the highest expression levels (Figure 3a). The myogenic genes *myf5* presented an inverse expression pattern compared with *pax7*, with the highest expression in the egg stage, which constantly decreased until the juvenile stage (Figure 3b). The two paralogues *myod1* and *myod2* presented differential expression profiles (Figure 3c,d). The *myod1* gene presented an intervallic increasing profile, while *myod2* showed maximum levels at 14 DPH, which decreased at the juvenile stage (Figure 3c,d). Our findings revealed that myogenin expression slightly decreased during ontogenesis from the egg stage to 10 DPH. Then, it presented a fast expression peak at 21 DPH, significantly decreasing afterward, reaching a low expression in the juveniles (Figure 3e).

Regarding *mrf4*, it presented an important up-regulation at 1 DPH, which kept medium expression values until 14 DPH, where the highest values were observed, along with a drastic decrease in the juveniles (Figure 3f). *Mymk* presented an inverse expression pattern compared with *pax7*, with high expression at the egg stage and 1 DPH. Then, the expression decreases at 3 DPH, followed by a stable level until 21 DPH. Afterward, the expression decreased to a very low level at the juvenile stage (Figure 3g). The expression of *mstnb* significantly peaked at 1 DPH, then the expression declined at 3 DPH and 8 DPH, rising again at 14 and 21 DPH (Figure 3h), and decreasing also in the juveniles.

### 3.3. Bone Responses during Ontogenesis

Regarding the osteogenesis-related genes, *col1a1a* had a highly similar expression pattern to *pax7* in muscle. *col1a1a* presented a low basal expression in the egg stage, which slightly increased from 1 DPH to 14 DPH; after that, an important up-regulation showed at 21 DPH and in the juveniles (Figure 4a). On the other hand, *fib1a* had a high expression in eggs, which progressively decreased, keeping lower expression from 3 DPH onward but with sporadic expression peaks at 5 and 14 DPH (Figure 4b). Concerning the osteogenic factor, the *on* highest level was found at the juvenile stage, although high levels were also observed at the egg and 1 DPH, significantly decreasing at 3 and 8 DPH and until 21 DPH (Figure 4c). Differently, the *op* expression was not significantly changed during the ontogenesis (Figure 4d). Finally, the expression of *ostc* was significantly low during the first 14 DPH and then increased at 21 DPH and in the juvenile (Figure 4e).

### 3.4. PLS-DA for Pattern and Biomarker Identification

In order to evaluate how many different larval stages can be properly identified based on the genomic data obtained from the gene expression analysis, a supervised multivariate classification analysis was performed. The selected method was a three-dimensional partial least squares (PLS) regression (a.k.a. projection on latent structures) discriminant analysis. The PLS-DA overview is shown in Figure 5A, highlighting that the results of the analysis were able to explain 43.8% of the total variability between samples. The specific percentages were Component 1 (22.1%), Component 2 (14.9%), and Component 3 (6.8%). The performance and accuracy of the analysis are shown in Figure 5B, which shows that R2 and Q2 (which are parameters that measure the internal and external predictability, respectively) are close to 0.8, indicating a relatively strong robustness of the analysis. The scatter 3D score plot of the PLS-DA is represented in Figure 6. In the plot, 1 and 2 allow a clear separation of four major clusters, from left to right: 0 day (egg); 1, 3, 5, 8, 10, and 14 days; 21 days; and juveniles. Among the second cluster, the 10 and 14 days’ samples are those that can be more clearly separated from the rest of the not separated time points, indicating that this stage can represent an inflection point in the ontogenic process of this species. The samples egg, 21 days, and juveniles are clearly different from others. So, according to obtained results, the pikeperch ontogeny can be distinguished into 5 stages: egg, days 1 to 8 (preflexion), days 10 to 14 (postflexion), day 21, and juveniles. In addition to identifying the larval stages, the results of the PLS-DA provide insights into which genes were most important to achieve this classification by assigning a Variable Importance in the Projection (VIP) score to each gene. This score summarizes the contribution of each variable to the model. Figure 7 shows the main VIP scores corresponding to each component, and the colored boxes on the right of each graph indicate the relative expression of the corresponding gene in each group. Only VIPs ≥ 1 are considered as relevant for accurate prediction and robustness. In all three components, the most relevant VIPs are *myf5* and *mymk* as best markers of earlier stages and *igf1ra*, *ostc*, *pax7*, and *ghr* as markers of later stages of ontogeny. The other VIPs were lower than one and did not pass the cutoff.

## 4. Discussion

The study of the expression of those genes related to fish somatic growth can provide new insights into fisheries management, aquaculture, and the development regulation during ontogenesis. The current study evaluated muscle and skeletal genes involved in the early steps of pikeperch ontogenesis to improve the basic knowledge of the musculoskeletal system during rearing and the associated challenges.

The water temperature was found to vary from 12.5 °C at hatching to 27.5 °C during the pikeperch study period. These changes within the different developmental stages of pikeperch during ontogeny provide a more targeted approach for assessing the effects of temperature changes on this species in both wild and laboratory conditions. So, the dynamics of gene expression in this study can be influenced by the changes in water temperature.

Growth hormone is one of the most important indicators for development and growth in vertebrates [18,30,31]. *gh* and *igfs* play an effective role in the growth of body cells and basal metabolism in fish [7,18,32]. Concerning *ghr*, it significantly increased at 1 DPH and tended to decrease at 21 DPH and in the juveniles. *ghr*, a single transmembrane receptor, is a vital factor regulating the *gh*/*igf* axis in fish [33]. In pikeperch, *igfI* was strongly upregulated after hatching, and from 10 to 14 DPH, the expression of *igfI* increased again; then, it returns to the lowest levels in the juveniles. This decrease in *igfI* in juveniles can stop muscle proliferation and development. Teng et al. [34] reported that *igfI* correlates more with fish growth. The initial feeding rate rapidly increased in pikeperch from 9 to 14 DPH and reached the highest level at 14 DPH [35]. According to our result, the expression levels of the *igfII* gene in egg and larvae stages were significantly different. Although a significant increase was observed at 14 DPH, the higher gene expression in the egg suggests that the *igfII* gene played an essential regulatory role in embryo and larvae development. Schäfer et al. [7] indicated that the transcript level of *igfII* and *ghr* in pikeperch increased from early after hatching to 4 and 7 DPH, and the highest levels at the juvenile stage may reflect the strong phase of growth in fingerlings. A previous report documented increased *igfII* transcript levels in maraena white fish (*Coregonus maraena*) at the onset of oral feeding and during development into the fingerlings [36]. In turbot (*Scophthalmus maximus*), the mRNA levels of *igfI* and *igfII* sharply increased from the stage of unfertilized egg to postlarvae, followed by a decrease with larval development [37]. These data indicate that both *igfI* and *igfII* are present in embryos (fertilized eggs), larvae, and the juvenile of pikeperch. However, in some specific points, such as before hatching, *igfII* seems to have a more notorious role than *igfI*, which markedly increased after hatching, and the expression profiles of both *igf*s were very similar. What they specifically do during the early life of fish is not sure, but it refers to a particular distribution of functions between both *igfs* in the entire life of the pikeperch. This is in agreement with the studies in gilthead sea bream (*Sparus aurata*), where *igfII* was more important than *igfI* in regulating myocyte proliferation [38,39].

During the ontogeny, *igf1bp4* showed a profile parallel to both *igfs* peptides, while *igf1bp5* fluctuated during development. These genes can control *igf* availability and activity in an autocrine and/or paracrine manner [40]. *Igfbp4* and *igfbp5* genes play an anabolic role in different species [31,41,42] and have also been proposed as a myogenic molecule promotor [9].

We showed that the gene expression of *igfrs* was different during ontogeny. Thus, *igf1ra* was highly expressed at 21 DPH and in the juveniles, while *igf1rb* was expressed from 1 DPH to 14 DPH. It suggests the different functions of both *igf1r* in pikeperch development, with *igf1ra* being more responsible for the earlier steps in the ontogeny in this species. Such differential roles of function of both *igf1r*s have been observed in other fish species, such as rainbow trout *Oncorhynchus mykiss* [43] and Chilean flounder *Paralichthys adspersus* [44]. In the current study, *pax7* expression, as a key marker of secondary myogenesis, started to increase importantly from 21 DPH to the juvenile stage, which has been shown in other fish species such as Japanese pufferfish (*Takifugu rubripes*) [45]. Of note, *pax7* is involved in developing the thymus, CNS, Shwann cells, cranial bone and cartilage, melanotrope cells, spermatogonia cells, and muscle [46]. The role of *pax7* in the late stages (21 DPH larvae to juvenile fish) is possible to correlate the gene expression levels with ontogenetic processes such as muscle differentiation, stratified and mosaic hyperplasia, or satellite cell proliferation [47,48,49].

The information on *mrfs* during larval development in pikeperch is scarce, and the results obtained can be helpful to understanding muscle growth regulation during ontogenesis and aquaculture development. Based on the current results, a set of *mrfs* were expressed during larval development. The two *myod* genes were differently expressed during development. The expression of *myod1* and *myod2* demonstrated that they exhibited overlapping but distinct patterns of expression in embryos and in juveniles. According to Franz et al. [14], the expression of *myod1* coincides with somitogenesis and the phases of muscle growth before and after hatching. The different expression patterns of *myod1* and *myod2* genes have been reported in sea bream, rainbow trout, and flounder (*Paralichthys olivaceus*) fast and slow muscles during embryos and adults [50].

The present study demonstrates that the expression of *myf5* and *mymk* was maximum before hatching and gradually decreased during the first stages of development, almost disappearing in the juveniles. It has been reported that *mymk* is involved in initial myoblast fusion during early development [51]. Both *myf5* and *mymk* are necessary for the initiation of myogenesis in vertebrates. In *Labeo rohita*, the expression of *myf5* was higher in the embryonic stages [52]. Concerning *myogenin*, this gene maintained relatively high expression levels in all the stages, with a peak at 21 DPH. *Myogenin* expression usually happens after *myf5* expression and remains active to maintain differentiation and growth of skeletal muscle fibers in the terminal differentiation stage [13]. Our results clearly showed that *mymk* and *myf5* were expressed at a high level in the early embryonic stages, whereas *mrf4* and myogenin were expressed after *mymk* and *myf5* expression with the peak at 14 or 21 DPH, which coincides with the differentiation and growth of skeletal muscle fibers of pikeperch.

Regarding the level of *mstnb* expression in pikeperch, results showed an initial rise at 1 DPH, when somitogenesis and muscle cell proliferation and differentiation are completed [53,54,55]. Thereafter, *mstnb* mRNA concentration declined to gradually reach a new peak at 14 and 21 DPH. According to Vianello et al. [55], *mstn* has a key regulatory role during the early muscle development in fish and mammals. Rodgers et al. [56] suggested that in developing tilapia (*Oreochromis mossambicus*), *mstn* was expressed early after hatching, as we observed in our results. In agreement with our study, the reports on zebrafish (*Danio rerio*) by Vianello et al. [55] and on pikeperch by Franz et al. [14] revealed high mstn expression in the juveniles.

The present study analyzed the expression profile of several gene markers for skeletogenesis during larval development in pikeperch. The high expression level of the *col1a1a* gene was found in 21 DPH and the juvenile stage. Durán et al. [57] and Gistelinck et al. [58] reported the expression of *col1a1a* within the formation of actinotrichia in the first fin skeleton during the development. However, in the present study, the lowest expression of the *col1a1a* genes was observed in pikeperch at early developmental stages. It can suggest the role of maternal collagen type I transcripts in the earliest stages of development [58]. Regarding the *fib1a* gene, the highest expression levels were observed in eggs. *fib1a* is produced by osteoblasts and accumulates in the ECM, which appeared necessary in the first stages of bone formation [59]. Its expression is higher during the early stages of osteoblast differentiation and declines during cell maturation [60]. According to Stein et al. [60] and Owen et al. [61], *fib1a* expression was highest on day 5. It progressively decreased after that, confirming a potential role in the first stages of osteogenesis and cell adhesion.

Concerning *on* and *ostc*, transcript levels were increased at 21 DPH and in the juveniles, confirming their essential role in establishing ECM production [62,63,64]. In agreement with these data, several studies demonstrated that the expression of *on* and *ostc* usually increases during maturation-specific skeletal cell types [62,64,65].

The pattern of gene expression during fish ontogeny often comprises more genes, making it challenging to interpret. PLS-DA helps reduce this dimensionality while retaining the important variation related to the developmental stages, facilitating the identification of key genes involved in ontogeny, which enables researchers to identify unique gene expression patterns associated with each stage of development.

This method provides visual output (e.g., score plots) that allows researchers to visualize the relationships and differences between various groups (e.g., showing the discrimination between the clustered samples corresponding to the different ontogenic stages) in a clear and interpretable way.

In summary, PLS-DA is a valuable analytical tool in the study of gene expression during the ontogeny of fish larvae, enabling researchers to derive important insights into developmental biology and related applications. The obtained results suggest that during the larval development stage of pikeperch, there is a complex and interesting pattern of gene expression that plays a crucial role in coordinating the proper development of the musculoskeletal system.

## 5. Conclusions

Overall, in this study, we investigated the mRNA expression of key genes involved in growth hormone regulation, muscle development, and osteogenesis in pikeperch from hatching through the juvenile stage. This research highlights distinct phases of gene expression during different developmental stages, with specific genes showing peak expression at various time points. The findings suggest a complex regulatory network influencing metabolism, cellular growth, and tissue development in pikeperch, establishing a basis for increasing knowledge and insight into how larvae and juvenile of an economically valuable species grow and develop. This knowledge holds great potential for informing and improving pikeperch aquaculture practices.

## Figures and Tables

**Figure 1 animals-14-03089-f001:**
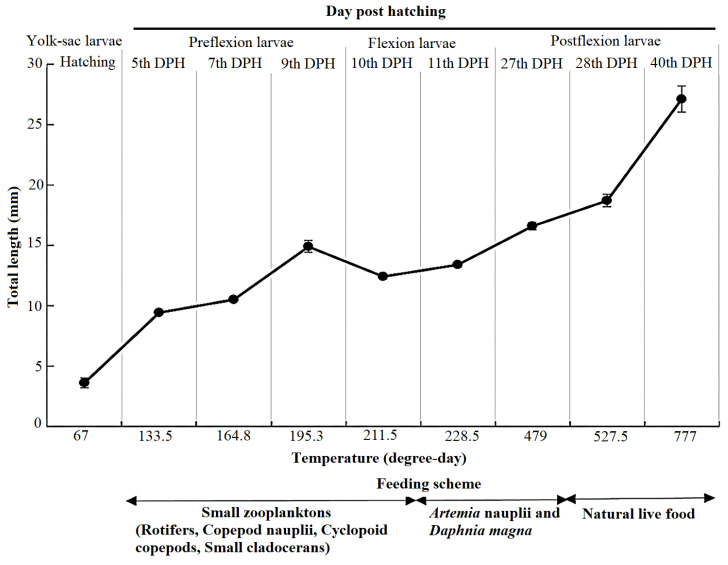
Feeding scheme for rearing of pikeperch (*Sander lucioperca*) and water temperature fluctuations during the first 40 days post-hatch.

**Figure 2 animals-14-03089-f002:**
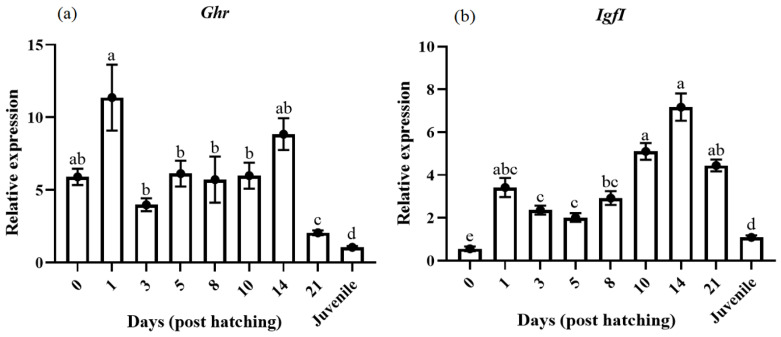
Relative gene expression of skeletal muscle *ghr* (**a**), *igfI* (**b**), *igfII* (**c**), *igf1bp4* (**d**), *igf1bp5* (**e**), *igf1ra* (**f**), and *igf1rb* (**g**) in pikeperch during the ontogenesis. Data are shown as means ± SEM (n = 10). Letters indicate significant differences (*p <* 0.05) by one-way ANOVA and Tukey HSD test. Sample 0 pointed to the sample egg before the hatch.

**Figure 3 animals-14-03089-f003:**
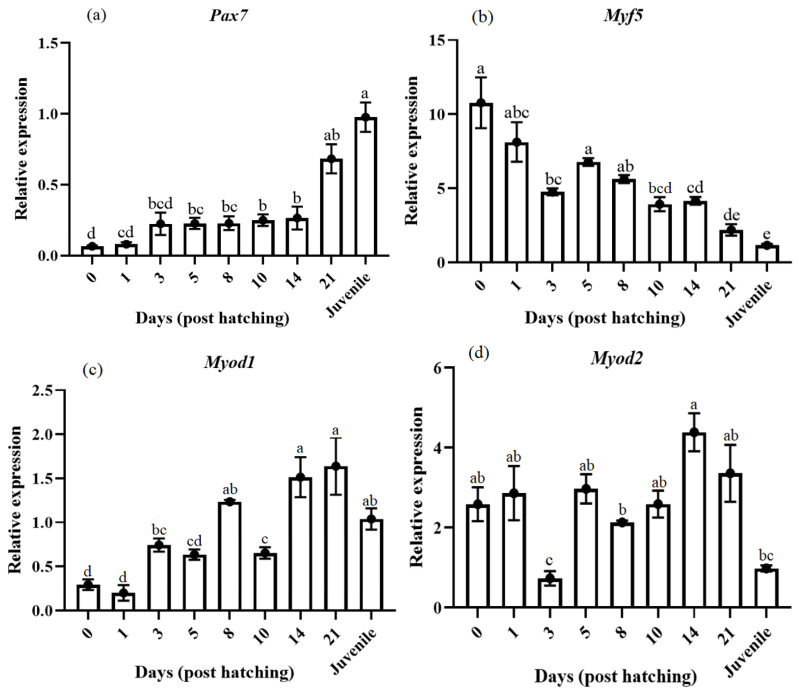
Relative gene expression of skeletal muscle *pax7* (**a**), *myf5* (**b**), *myod1* (**c**), *myod2* (**d**), *myogenin* (**e**), *mrf4* (**f**), *mymk* (**g**), and *mstnb* (**h**) in pikeperch during the ontogenesis. Data are shown as means ± SEM (n = 10). Letters indicate significant differences (*p <* 0.05) by one-way ANOVA and Tukey HSD test. Sample 0 indicated to the sample egg before hatch.

**Figure 4 animals-14-03089-f004:**
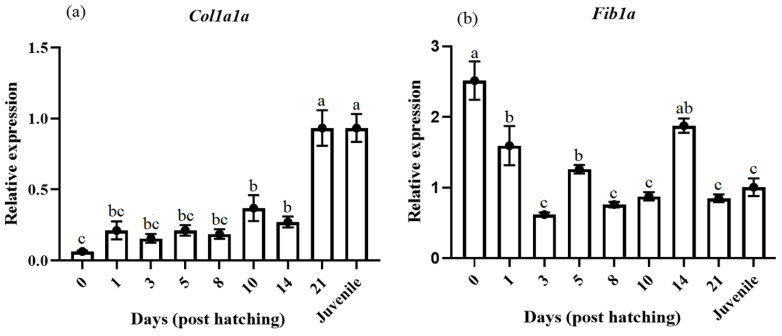
Relative gene expression of bone *col1a1a* (**a**), *fib1a* (**b**), *on* (**c**), *op* (**d**), and *ostn* (**e**) in pikeperch during the ontogenesis. Data are shown as means ± SEM (n = 10). Letters indicate significant differences (*p <* 0.05) by one-way ANOVA and Tukey HSD test. Sample 0 pointed to the sample egg before the hatch.

**Figure 5 animals-14-03089-f005:**
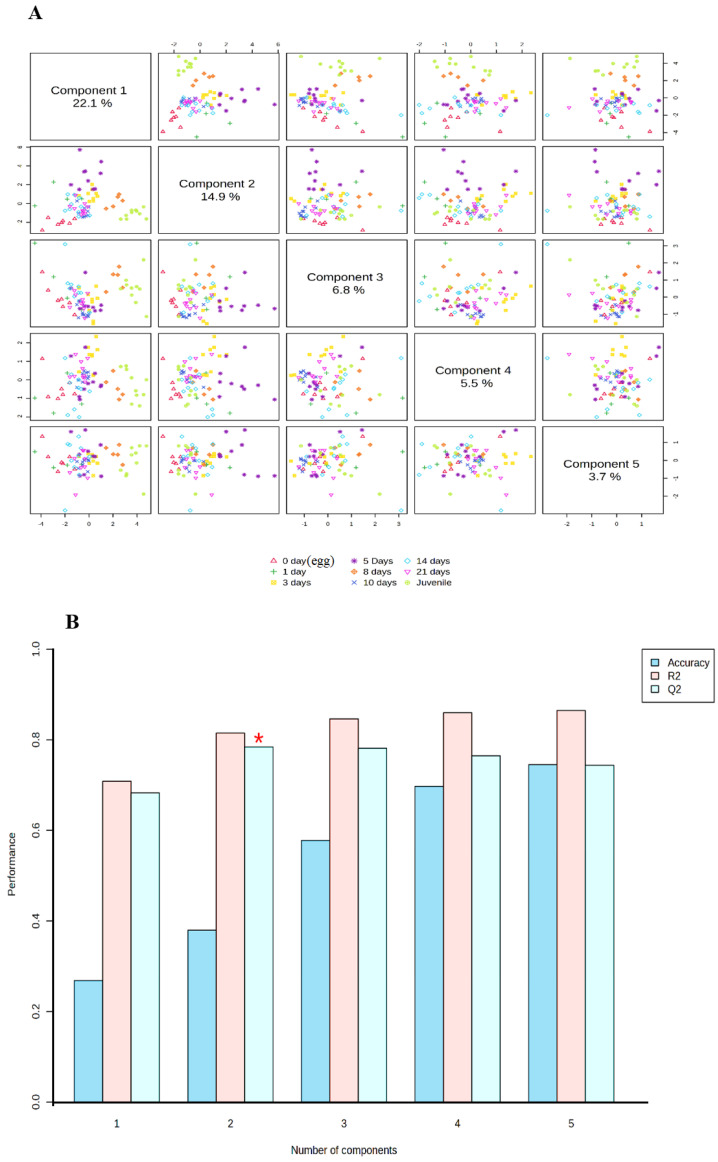
Overview of the PLS-DA model with the % of variability explained by each component (**A**) and the quality assessment of the model with the R2 and Q2 values (**B**). The asterisk indicates which number of components combined gives a higher Q2 value.

**Figure 6 animals-14-03089-f006:**
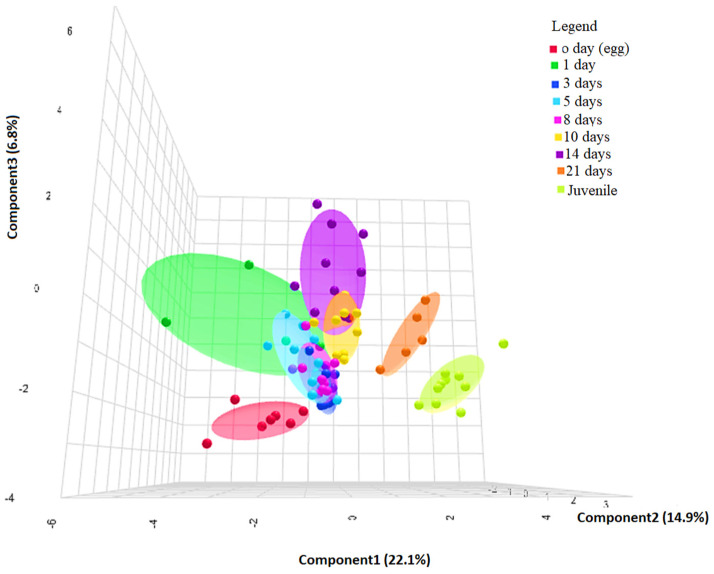
Scatter 3D score plot of the PLS-DA shows the discrimination between the clustered samples corresponding to the different ontogenic stages of *Sander lucioperca*. The position of each dot along the axis indicates the strength and direction of the relationship between the variable and the principal component.

**Figure 7 animals-14-03089-f007:**
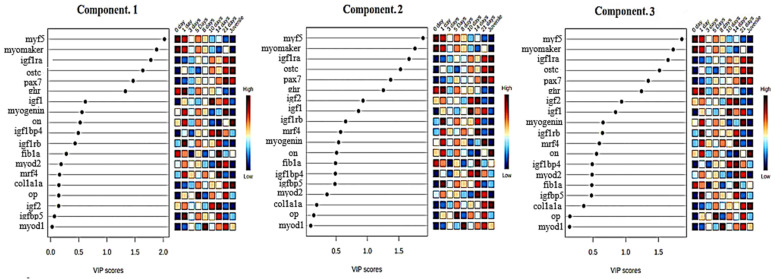
Overview of the main variable importance in projection (VIP) corresponding to the three main components (Comp), and the colored boxes on the right of each graph indicate the relative expression of the corresponding gene in each group. Only VIPs ≥ 1 are considered relevant for accurate prediction and robustness of the model. The colored boxes on the right of each graph indicate the relative expression of the corresponding gene in each group. 0 day (egg).

**Table 1 animals-14-03089-t001:** Overview of selected genes and primers for pikeperch qPCR analysis.

Gene Symbol	Sense Primer (5′-3′)	Annealing Temperature (°C)	Amplification Efficiency (%)	Accession Number
Reference genes				
*βactin*	F: CGACATCCGTAAGGACCTGT	55	92.9	XM_031320710.2
	R: GCTGGAAGGTGGACAGAGAG			
*Ef1a*	F: TGATGACACCAACAGCCACT	55	90.1	XM_031295658.2
	R: AAGATTGACCGTCGTTCTGG			
*rps18*	F: TGACGGAAGGGCACCACCAG R: AATCGCTCCACCAACTAAGAACGG	60	91.7	XR_004103451.1
Target genes				
Growth genes				
*Ghr*	F: GGAGAGCACCTTAGCCACAGAG R: GTGGTGTAGCCGCTTCCTTCT	58	99.6	XM_031305905.2
*IgfI*	F: CTGTGCACCTGCCAAGACTA R: TGTGCCCTTGTCCACTTTGT	60	92.8	XM_031313442.2
*IgfII*	F: AAGACACGGACACCACTCAC R: TGCCGAGGCTATTTCCACAG	58	94.3	XM_031282902.2
*Igf1bp4*	F: CCATTTAATCCAGCCCCCGA R: GCTATGGCACATGAAACGGC	58	95.9	XM_031284490.2
*Igf1bp5*	F: CCATTTAATCCAGCCCCCGA R: GCTATGGCACATGAAACGGC	55	104.6	XR_004896646.1
*Igf1ra*	F: ACATTCATTCACCCCTCCCCT R: ATACTCGGACCACAGATCTCTCC	54	97.1	XM_031293486.2
*Igf1rb*	F: ATAACTGCCCCTTACCCTGTC R: TGACGGTGAGGTTGGGAAAG	56	95.7	
Muscle genes				
*Pax7*	F: GTCCCTTCAGGTGAGGCTTCC R: GACTCCACGTCTGACACTTCA	55	95.9	XM_031301093.2
*Myf5*	F: CCAGTGTGGCACCATCTGAA R: GTTGCCTAAACTGTCGTTCCTC	56	96.2	XM_031313525.2
*Myod1*	F: CAAACGGCGGTCTGAAGAGT R: GTGTGAGTGGACGGTTAGGC	58	96.4	XM_031279879.1
*Myod2*	F: GGGATGACGGCTTCTACTCG R: TTCGGGTTGAACACGGAGAG	57	97.1	XM_031323772.2
*Myogenin*	F: CCCAGCCCAGAGTGTCGTC	58	96.7	XM_031323467.2
	R: GTGCACATATGGGTCCGCTG			
*Mrf4*	F: AAAGTCAGCCCCGACGGATA R: CTCCACCTTGGGTAGCCTCT	57	98.7	XM_031313716.2
*Mymk*	F: ATGTTCTTCACTGCGATCTACC	55	96.1	XM_031305655.2
	R: AGAGAGCTGTGCCGTAAACAC			
*Mstnb*	F: TCTGTCCTCCCGCCTTATGA R: TTTCCCTTTGTGCTGGTCGT	58	101.2	XM_031304808.2
Bone genes				
*Col1A1A*	F: GGAGGGACTTAAAGGAAACCGT R: CTCACGTCCTGGCTCACCG	58	83.6	XM_031292502.2
*Fib1a*	F: CACAAACCACTGCCCCTGAT	54	85.8	XM_031281780.2
	R: AGTCACAGATGTTGCCGTGT			
*On*	F: CGACACCTCTTGCCAGTTCT	56	97.2	XM_031317122.2
	R: CTCATACGCAGGGGGAACTC			
*Op*	F: CAGTGTACAAGGTAAAGGCTCT	52	91.2	XM_031277598.2
	R: CTGAGTTCCTGGTCCTGAGA			
*Ostc*	F: GGCTGTCGTCTGTCTGACTTC R: CTCTCCACAAACAAACCCTCCTG	56	93.3	XM_035996639.1

## Data Availability

The data that support the findings of this study are available from the corresponding author upon reasonable request.

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
