# Peer review of "The Pattern of Gene Expression (Igf Family, Muscle Growth Regulatory Factors, and Osteogenesis-Related Genes) Involved in the Growth of Skeletal Muscle in Pikeperch (Sander lucioperca) During Ontogenesis"

_animals, 2024, doi:10.3390/ani14213089_

Round 1

Reviewer 1 Report

Comments and Suggestions for Authors

The manuscript by Lavajoo et al. titled "The Pattern of Gene Expression (Igf Family, Muscle Growth Regulatory Factors, and Osteogenesis-Related Genes) Involved in the Growth of Skeletal Muscle in Pikeperch (Sander lucioperca) during Ontogenesis" focused on the mRNA expression of various growth-related and osteogenesis genes in pikeperch (Sander lucioperca) during its developmental stages. Results showed distinct phases of gene expression across the egg, larval, and juvenile stages, indicating a transition towards juvenile development. these findings highlight the dynamic changes in gene expression that regulate metabolism, growth, and development in pikeperch, providing insights important for pikeperch farming. This work is various and substantial. However, there are some mistakes in this manuscript. I suggest that it is worth to be published in this journal after modified. 

1.Some gene abbreviations in the article are inconsistent, please verify!

2.The quality of Figure 6 needs to be improved.

3. There are formatting issues in the article, such as many places not being italicized.

Comments on the Quality of English Language

Moderate editing of English language required.

Author Response

For research article

Responding to Reviewer Comments

1. Summary        

Thank you very much for taking the time to review our manuscript ''The Pattern of Gene Expression (Igf Family, Muscle Growth Regulatory Factors, and Osteogenesis-Related Genes) Involved in the Growth of Skeletal Muscle in Pikeperch (Sander lucioperca) during Ontogenesis''. 

We have made revisions in all parts of the manuscript based on the comments. Typographical and grammatical errors in the manuscript has been checked.
Thank you for the very good, constructive comments and suggestions. We feel that incorporating their feedback has greatly improved the manuscript overall.

Best regards,
Joaquim Gutiérrez
Department of Cell Biology, Physiology and Immunology,
Faculty of Biology, 
University of Barcelona, Spain

2. Questions for General Evaluation    Reviewer’s Evaluation    Response and Revisions
Does the introduction provide sufficient background and include all relevant references?    Yes/Can be improved/Must be improved/Not applicable    
Are all the cited references relevant to the research?    Yes/Can be improved/Must be improved/Not applicable    
Is the research design appropriate?    Yes/Can be improved/Must be improved/Not applicable    
Are the methods adequately described?    Yes/Can be improved/Must be improved/Not applicable    
Are the results clearly presented?    Yes/Can be improved/Must be improved/Not applicable    
Are the conclusions supported by the results?    Yes/Can be improved/Must be improved/Not applicable    

3. Point-by-point response to Comments and Suggestions for Authors

Reviewer 1
Comments 1: [Some gene abbreviations in the article are inconsistent, please verify!]
Response 1: 
Many thanks for your comment. We agree with this comment. Therefore, It has been corrected in the text. In the revised manuscript lines 16, 34, 258, 315: mymk corrected; line 182: s18 deleted and line 252: myognin is italicized.

Comments 2: [The quality of Figure 6 needs to be improved.]
Response 2: Many thanks for your comment. Agree. We have, improved and changed legend section, and component 1, 2 and 3 in figure 6 and new figure 6 inserted. Page 16/22

Comments 3: [There are formatting issues in the article, such as many places not being italicized.]
Response 3: Many thanks for your comment. Agree. All the text checked and typographical errors corrected.

 Reviewer 2:
Comments 1: [Suggestions for the text are made in the attached document].
Response 1: Many thanks for your comments. We agree with all these comments. It has been corrected in the text. All changes are in the revised manuscript.

Comments 2: [The methodology could have another order.]
Response 2: Many thanks for your suggestion. Agree. The methodology was changed based on the mentioned suggestion. 2.1. Larval and Juvenile Rearing; 2.2. Ethics Approval; 2.3. Gene Selection and Primer Design; 2.4. RNA Extraction, cDNA Synthesis, and qPCR analysis; 2.5 Statistical Analyses. The total length measurement sentence has been inserted in the revised manuscript. After total length measurement, the whole fish was immediately frozen in liquid nitrogen (line 132).

 Reviewer 3
Comments 1: [ In table 1, please add the genes ncbi accession numbers].
Response 1: Many thanks for your comments. We agree with this comment. It has been inserted in the table 1 with new column Accession number. 

Comments 2: [Please correct figure 4 because it looks cropped.]
Response 2: Many thanks for your notice and comment. Agree. In Figure 4 the Ostc graph has been corrected.

Comments 3: [Please add some more details either in the results or in discussion section for the PLS-DA analysis. What more information it offers? In my understanding it’s added value is not part of the discussion.]
Response 3: Many thanks for your suggestion and comment. We agree with this comments. 
In order to evaluate how many different larval stages could be properly identified based on the genomic data obtained from the gene expression analysis, a supervised multivariate classification analysis was performed. The selected method was a three dimensional partial least squares (PLS) regression (a.k.a. projection on latent structures) discriminant analysis. The PLS-DA overview ….

In addition to identifying the larval stages, the PLS-DA analysis provides insights into which genes were most important to achieve this classification by assigning a Variable Importance in the Projection (VIP) score to each gene. This score summarizes the contribution of each variable to the model. These two paragraphs has been inserted in Result in section 3.4 (Lines 290-294 and lines 307-311.
And 
The pattern of gene expression during fish ontogeny often comprises more genes, making it challenging to interpret. PLS-DA helps reduce this dimensionality while retaining the important variation related to the developmental stages, facilitating the identification of key genes involved in ontogeny., which enabling researchers to identify unique gene expression patterns associated with each stage of development.
This method provides visual output (e.g., score plots) that allows researchers to visualize the relationships and differences between various groups (e.g., showing the discrimination between the clustered samples corresponding to the different ontogenic stages) in a clear and interpretable way.
In summary, PLS-DA is a valuable analytical tool in the study of gene expression during the ontogeny of fish larvae, enabling researchers to derive important insights into developmental biology and related applications.
These three paragraphs has been inserted in discussion section (Lines 444-455).

Comments 4: [In page 15, line 323: do you mean decreased instead of increased in the beginning of the sentence?.]
Response 4: Many thanks for your comment. The initial feeding rate increased rapidly in pikeperch from 9 to 14 DPH and reached the highest level at 14 DPH (line 355-356).
Comments 5:
 Response 5: Many thanks for your comments on the English Language. Typographical and grammatical errors in the manuscript has been corrected in the revised version as you mentioned. We feel that these comments has greatly improved the manuscript overall.

4. Response to Comments on the Quality of English Language
Point 1:
Response 1: Typographical and grammatical errors in the manuscript has been checked and corrected in the revised version. 

5. Additional clarifications
The methodology order section 2.3 and 2.4 has been changed. As a clarity some new sentences and paragraphs added to the methodology, results and discussion sections. In Figure 1, border deleted. In figure 4, the Ostc is corrected due to one part being cropped. The quality of figure 6 has been improved. In table 1 Accession number column inserted. Some gene abbreviations has been checked and corrected. Reference section checked and corrected. Typographical and grammatical errors in the manuscript has been checked and corrected in the revised version.
The below paragraph has been inserted in Methodology section 2.4 (lines 164-167).
Due to the different amount of sample available, the number of sample for each stage was the following: 0 day (egg): 10 samples; day 1: 5 samples; day 3: 10 samples; day 5: 10 samples; day 8: 10 samples; day 14: 10 samples; day 21: 6 samples; Juveniles: 10 samples.

Reviewer 2 Report

Comments and Suggestions for Authors

-Suggestions for the text are made in the attached document.

-The methodology could have another order:

2.1. Larval and Juvenile Rearing

2.2. Ethics Approval

2.3. Gene Selection and Primer Design

2.4. RNA Extraction, cDNA Synthesis, and qPCR analysis (or create a section just for qPCR analysis)

2.5 Statistical Analyses

-The methodology does not mention that the total length of the larvae is taken.

Comments on the Quality of English Language

The authors have some errors in the English that could be fixed. I have indicated most of them in my review.

Author Response

(The authors gave the same response as above.)

Reviewer 3 Report

Comments and Suggestions for Authors

Dear Editor,

The manuscript entitled “The Pattern of Gene Expression (Igf Family, Muscle Growth Regulatory Factors, and Osteogenesis-Related Genes) Involved in the Growth of Skeletal Muscle in Pikeperch (Sander lucioperca) during Ontogenesis” by Fatemeh Lavajoo et al. presents a study focused on the mRNA expression analysis of the growth hormone (gh)/insulin-like growth factor (igf) axis (ghr, igfI, igfbp, igfr), muscle regulatory factors (pax7, myf5, myod, myogenin, mrf, mymk, mstn) and osteogenesis related genes (colla1a, fib1a, on, op, ostn) from hatching through day 40th post hatching (DPH), for the pikeperch (Sander lucioperca). The authors observed and commented three phases of gene expression in the day 0 (egg), larval, and juvenile stages of pikeperch, which can be a progression or transition from the initial state toward the juvenile state.

Τhe manuscripts’ objects are interesting, it well written in a comprehensive way and the findings are interesting and justified.  Therefore, the manuscript could be accepted for publication after some minor revisions:

1.      In table 1, please add the genes ncbi accession numbers

2.      Please correct figure 4 because it looks cropped

3.      Please add some more details either in the results or in discussion section for the PLS-DA analysis. What more information it offers? In my understanding it’s added value is not part of the discussion.

4.      In page 15, line 323: do you mean decreased instead of increased in the beginning of the sentence?

Author Response

(The authors gave the same response as above.)
